# Repurposed Drugs That Activate Autophagy in Filarial Worms Act as Effective Macrofilaricides

**DOI:** 10.3390/pharmaceutics16020256

**Published:** 2024-02-09

**Authors:** Denis Voronin, Nancy Tricoche, Ricardo Peguero, Anna Maria Kaminska, Elodie Ghedin, Judy A. Sakanari, Sara Lustigman

**Affiliations:** 1Systems Genomics Section, Laboratory of Parasitic Diseases, Division of Intramural Research, NIAID, NIH, Bethesda, MD 20892, USA; elodie.ghedin@nih.gov; 2Molecular Parasitology, New York Blood Center, Lindsley F. Kimball Research Institute, New York, NY 10065, USA; 3Department of Pharmaceutical Chemistry, University of California, San Francisco, CA 94158, USA; judy.sakanari@ucsf.edu

**Keywords:** *Wolbachia*, autophagy, repurposed drugs, filarial diseases, macrofilaricidal drugs

## Abstract

Onchocerciasis and lymphatic filariasis are two neglected tropical diseases caused by filarial nematodes that utilize insect vectors for transmission to their human hosts. Current control strategies are based on annual or biannual mass drug administration (MDA) of the drugs Ivermectin or Ivermectin plus Albendazole, respectively. These drug regimens kill the first-stage larvae of filarial worms (i.e., microfilariae) and interrupt the transmission of infections. MDA programs for these microfilaricidal drugs must be given over the lifetime of the filarial adult worms, which can reach 15 years in the case of *Onchocerca volvulus*. This is problematic because of suboptimal responses to ivermectin in various endemic regions and inefficient reduction of transmission even after decades of MDA. There is an urgent need for the development of novel alternative treatments to support the 2030 elimination goals of onchocerciasis and lymphatic filariasis. One successful approach has been to target *Wolbachia*, obligatory endosymbiotic bacteria on which filarial worms are dependent for their survival and reproduction within the human host. A 4–6-week antibiotic therapy with doxycycline, for example, resulted in the loss of *Wolbachia* that subsequently led to extensive apoptosis of somatic cells, germline, embryos, and microfilariae, as well as inhibition of fourth-stage larval development. However, this long-course regimen has limited use in MDA programs. As an alternative approach to the use of bacteriostatic antibiotics, in this study, we focused on autophagy-inducing compounds, which we hypothesized could disturb various pathways involved in the interdependency between *Wolbachia* and filarial worms. We demonstrated that several such compounds, including Niclosamide, an FDA-approved drug, Niclosamide ethanolamine (NEN), and Rottlerin, a natural product derived from Kamala trees, significantly reduced the levels of *Wolbachia* in vitro. Moreover, when these compounds were used in vivo to treat *Brugia pahangi*-infected gerbils, Niclosamide and NEN significantly decreased adult worm survival, reduced the release of microfilariae, and decreased embryonic development depending on the regimen and dose used. All three drugs given orally significantly reduced *Wolbachia* loads and induced an increase in levels of lysosome-associated membrane protein in worms from treated animals, suggesting that Niclosamide, NEN, and Rottlerin were effective in causing drug-induced autophagy in these filarial worms. These repurposed drugs provide a new avenue for the clearance of adult worms in filarial infections.

## 1. Introduction

Onchocerciasis (River Blindness) and lymphatic filariasis (Elephantiasis) are two major Neglected Tropical Diseases (NTDs) caused by *Onchocerca volvulus* and *Brugia malayi* or *Wuchereria bancrofti* infections, respectively, that, together, affect millions of people worldwide in mostly poor, developing countries [1]. To date, there are no vaccines to prevent these infections and no drugs that directly kill the adult stages (macrofilaricidal drugs) [2,3,4] that can be used in mass drug administration (MDA). Therefore, current international control programs attempt to interrupt the transmission of infection with microfilaricidal drugs that kill microfilariae (mf), the stage of the parasite that is transmitted from infected individuals to the respective vectors [1,2,5,6,7,8,9]. Efforts to interrupt transmission rely on annual or biannual MDA using Ivermectin, Moxidectin, Albendazole, and Diethylcarbamazine individually or in combination, depending on the target worm. Treatments are given over the lifetime of the adult worms (10–15 years for *O. volvulus*, 6–8 years for *Wuchereria* and *Brugia* spp.) [2,3,4,9,10,11]. However, the lack of macrofilaricidal drugs [11] makes it unlikely that the 2030 goals of onchocerciasis elimination will be reached [12,13,14]. To address these challenges and ultimately meet the elimination goals, new macrofilaricidal drugs are urgently needed [15,16,17]. One successful approach has been to target *Wolbachia*, obligatory endosymbiotic bacteria on which human filarial parasites are dependent [18,19,20,21,22,23]. In filarial nematodes, *Wolbachia* are essential for normal larval growth and development, embryogenesis, and survival of adult worms [24]. These processes have a high metabolic demand because of the rapid growth, development, and organogenesis of the various stages of the filarial parasites. They are also associated with the rapid expansion of *Wolbachia* populations following larval infection of mammalian hosts and in reproductively active adult female worms [25]. Moreover, loss of *Wolbachia* results in extensive apoptosis of germline and somatic cells, embryos, microfilariae, and inhibition of fourth-stage larval development, presumably because the provision of essential nutrients or metabolites required to prevent apoptosis of these cells and tissues is interrupted [26,27]. Thus, apoptosis due to the loss of *Wolbachia* accounts for some of the anti-filarial activities of antibiotic therapies.

Recent data demonstrate the unique potential of *Wolbachia* as an effective chemotherapeutic target against human filarial infections [1,24,28,29]. However, most anti-*Wolbachia* drugs are known antibiotics that have limitations when considered for use in MDA programs. A strategy that is focused on disturbing the symbiotic interplay between worms and bacteria could be an alternative approach to the use of bacteriostatic antibiotics. Indeed, previous studies have identified intracellular processes that are central to the symbiosis between the filarial host *Brugia* and its *Wolbachia*. Importantly, interference with these co-dependent processes (metabolism, biosynthesis, intracellular defense) has killed *Wolbachia* and, consequently, the filarial worms [26,27,30]. Moreover, the intracellular defense processes of eukaryotic cells also play an essential role in maintaining symbiosis. Particularly, autophagy was shown to regulate the size of the intracellular population of *Wolbachia* in different hosts [27,31,32]. Data suggest that *Wolbachia* interact with and potentially modify a key component of the autophagosome, ATG8, to ensure their survival [27], while the host utilizes autophagy to control the intracellular bacteria populations. An essential role of autophagy in symbiosis was shown for three functionally distinct *Wolbachia* strains: mutualistic (wBm and its host *B. malayi*), pathogenic (wMelPop in *Drosophila melanogaster*), and parasitic (*w*AlbB in C6/36 *Aedes albopictus* mosquito cells) [27,33]. These studies have concluded that the induction of an intracellular defense mechanism in filarial nematodes might be a novel, promising strategy to block *Wolbachia’s* beneficial function in symbiosis [27,33].

Building on the knowledge that *Wolbachia* populations are under direct and continuous control by autophagy, we tested compounds known to induce autophagy to determine whether, by accelerating these processes, we can indirectly affect both *Wolbachia* of *Brugia pahangi* (*w*Bp) and, consequently, its filarial host, *B. pahangi*. We hypothesized that compounds that target essential elements needed for maintaining this symbiotic relationship confer a distinct advantage over those that directly kill bacteria (antibiotics) or directly target the filarial worm. These drugs could, in principle, exploit the filarial host’s innate immunity to restrict the endosymbiont population, leading to disruption of worm fitness and survival. This proof-of-concept study clearly demonstrates that Niclosamide and Rottlerin, two drugs that are known to induce autophagy, significantly reduced the levels of *Wolbachia* in *B. pahangi* worms in vitro. When used in vivo to treat *B. pahangi*-infected gerbils, Niclosamide and Rottlerin were shown to also affect microfilariae secretion by female worms and embryogenesis; Niclosamide and Niclosamide ethanolamine (NEN) also impacted the survival of the adult worms in vivo. Future optimization studies will be necessary if *Wolbachia* are to be effectively exploited as a target for the successful treatment of onchocerciasis and lymphatic filariasis.

## 2. Material and Methods

### 2.1. Ethics Statement

Animal studies were approved under the New York Blood Center (NYBC) Institutional Animal Care and Use Committee Protocol 369 and adhered to the guidelines set forth in the NIH Guide for the Care and Use of Laboratory Animals and the USDA Animal Care Policies.

### 2.2. In Vitro Screening of Compounds with Adult Brugia pahangi Female Worms

Live adult female *Brugia pahangi* worms recovered from infected Mongolian gerbils (*Meriones unguiculatus*) about 120 days post-infection were provided by the NIH/NIAID Filariasis Research Reagent Resource Center. Upon arrival, individual adult female *B. pahangi* were placed in 2 mL complete media (RPMI-1640 supplemented with 25 mM L-Glutamax, 2.0 g/L NaHCO_3_, (Life Technologies, Grand Island, NY, USA), 10% heat-inactivated fetal bovine serum (FBS) (Hyclone, Logan, UT, USA), and 1× Pen-Strep, (Life Technologies, Grand Island, NY, USA) in 24-well plates. Compounds (Appendix A) were dissolved in 100% DMSO and added to each well at varying concentrations in 0.1 to 0.5% DMSO with at least 3–5 replicates per compound. Eleven to twenty-two replicates of 0.1 to 0.5% DMSO (depending on the final DMSO concentration in the tested compound) were used as vehicle control, and five replicates of 12.5 µM Doxycycline (Sigma, St. Louis, MO, USA) were used as a positive control (Appendix A). Cultures were maintained in a 37 °C incubator over the course of the 6-day assay. Motility measurements (Appendix A) were taken over days 2–6 in culture and estimated qualitatively on day 6 (relative score, 0%, 25%, 50%, 75%, and 100%). Secretion of microfilariae per well (Appendix A) was estimated qualitatively (relative score, 0%, 25%, 50%, 75%, and 100%) or quantitatively (total number of microfilariae secreted per worm) over the last day (day 5–6) in culture (*n* = 3–19 per condition). Moreover, in some treatment conditions, after 6 days in culture, 6 female worms were also analyzed for fitness of the developing embryonic stages within the gonads using the embryogram assay. The remaining female worms were frozen individually in an Eppendorf tube and kept at −80 °C until further DNA extraction and qPCR (Appendix A).

### 2.3. qPCR Analysis of Wolbachia in Adult Brugia pahangi

Adult worms collected from in vitro cultures or after necropsies were washed with PBS and snap-frozen in liquid nitrogen prior to storage at −80 °C. Genomic DNA (gDNA) was extracted from individual female worms using a DNEasy Blood & Tissue Kit (QIAGEN, Venlo, The Netherlands) according to the manufacturer’s instructions. Quantitative PCR (qPCR) was performed using a GeneCopoeia 2× All-in-One Master Mix (Cat #QP001- 01) in a Bio-Rad CFX Connect RT-PCR thermocycler. Primers for the single copy *Wolbachia* surface protein gene (*wsp*) were used to quantify *Wolbachia* loads, and the single copy glutathione- S-transferase gene (*gst*) was used to quantify *Brugia* genomes per worm following the protocol of McGarry et al. [25].

### 2.4. Western Blot LAMP Assay

*Brugia pahangi* female worms were collected from the in vitro treated and untreated wells or from worms recovered from treated and untreated gerbils. All samples were washed three times in cold PBS, and two worms were transferred to a tube with 50 µL of RIPA buffer (Invitrogen, Waltham, MA, USA) with 0.1% SDS to lyse the worm samples. Crude protein extracts of the worms were then mixed with the loading sample buffer (BioRad, Hercules, CA, USA), boiled, and run in 4–20% gradient SDS-PAGE gels (poly-acrylamide gel electrophoresis, Bio-Rad, Hercules, CA, USA). Proteins were transferred to nitrocellulose membranes and probed with anti-LAMP antibodies (Abcam, Waltham, MA, USA) and anti-tubulin (Sigma, St. Louis, MO, USA) antibodies by Western blot to detect LAMP (lysosome-associated membrane protein) and alpha-tubulin, followed by secondary antibodies anti-rabbit and anti-mouse antibodies, tagged with infrared fluorescent dye (Li-Cor Biosciences, Lincoln, NE, USA), respectively. Signals were observed and quantified using a Li-Cor instrument. Western blots were performed using three independent protein samples; fluorescent units of the LAMP signals were divided by fluorescent units of tubulin signals for the same sample. Relative expression of LAMP was compared between treated and control samples using a *t*-test.

### 2.5. Animal Infections and Treatment of Brugia pahangi-Infected Gerbils

Male gerbils 50–60 g, 5–7 weeks in age, were injected intraperitoneally (IP) with 200 third-stage larvae of *B. pahangi* (NIAID/NIH Filariasis Research Reagent Resource Center, University of Georgia, Athens, GA, USA), as previously described [34]. Infected gerbils, 5 months post-infection, were treated for 10 days as follows: (1) in the first pilot study, gerbils were treated with Niclosamide (Sigma, St. Louis, MO, USA) orally (PO, 200 mg/kg, *n* = 3) or intraperitoneally (IP, 10 mg/kg, *n* = 3). The drug was dissolved in 0.5% carboxymethyl cellulose (Sigma, St. Louis, MO, USA) and 2% Tween 20 in phosphate-buffered saline (PBS). The control group (*n* = 2) was treated with the vehicle solution only. (2) Gerbils were treated IP with 20 mg/kg with either Niclosamide (*n* = 3) or Niclosamide ethanolamine (NEN, Cayman Chemical Co., Ann Harbor, MI, USA) (*n* = 4) dissolved in 0.5% carboxymethyl cellulose and 2% Tween 20 in PBS. Gerbils in the control group (*n* = 3) were treated IP with vehicle solution only. (3) Gerbils were treated orally once a day PO with 200 mg/kg Niclosamide (*n* = 3) or 200 mg/kg NEN (*n* = 3) in 0.5% carboxymethyl cellulose and 2% Tween 20 in PBS. The Control group (*n* = 3, experiment 2) was treated IP with vehicle solution only. (4) Gerbils were treated orally two times a day (BID) with 100 mg/kg Niclosamide (*n* = 2) or 100 mg/kg NEN (*n* = 2) in 0.5% carboxymethyl cellulose and 2% Tween 20 in PBS. The Control group (*n* = 2) was treated BID with vehicle solution only. All the treated animals were necropsied 4–6 days post-last dose. In the last in vivo experiment, infected gerbils were treated orally BID with 50 mg/kg Rottlerin (Enzo Life Sciences, Farmingdale, NY, USA) (*n* = 6) or with vehicle alone (*n* = 9) as the control group for only 5 days. The treated animals were necropsied 3–6 days post-last dose. For all in vivo studies, as previously reported [15], worms were collected from the gerbil’s peritoneal cavity, counted, sexed, and examined under a dissecting microscope before being processed for analyses (Appendix A). To determine the effect of the compound on *Wolbachia* loads, individual female worms were frozen (*n* = 8–46, depending on the experiment and treatment group) and analyzed by qPCR. To determine the effect of the compound on worm fecundity, worms (*n* = 6–51, depending on the experiment and treatment group) were cultured individually, and the microfilariae secreted overnight from each female worm were collected and counted. Female worms after overnight cultures (*n* = 2–6 per gerbil) were processed for embryogram analysis, and 2–8 worms per gerbil were fixed for transmission electron microscopy (TEM).

### 2.6. Analysis of B. pahangi Adult Female Worm Fecundity Ex Vivo

To test the effects of the compounds on female worm fecundity, all surviving female worms were examined ex vivo to quantify the number of microfilariae (mf) released overnight in culture and to profile the various stages of embryonic development within the uteri of the individual female worms, as previously described [34]. After necropsy, recovered female worms were incubated in individual wells of a 24-well plate with 2 mL complete media at 37 °C and 5% CO_2_. After 24 h, the number of mf released by each adult female was determined by counting the number of mf in a 0.5 mL aliquot and multiplying by 4 to calculate the total number secreted by each worm. Additionally, embryogram analyses on the individual female worms were performed as described [34]. Briefly, each female worm was homogenized first in 0.5 mL PBS to release the uterine content. The suspension was then spun down, 0.4 mL of the PBS was removed, and the remaining 0.1 mL was used for analysis. The total gonad contents were determined in two 10 µL aliquots using a hemocytometer, while the distribution of the various embryonic developmental stages (eggs, embryos, pre-microfilariae, and stretched mf) was determined in two separate aliquots containing at least 200 events each on a 30-7H-Black slide using a compound microscope. A minimum of 200 events were assessed from each female worm and at least six females per treated gerbil. The intrauterine embryogram was expressed as the relative proportions of the different stages of development: eggs, embryos, pre-microfilariae, stretched mf, and deformed embryos in the total number of events recorded.

### 2.7. Statistical Analysis of the In Vivo Study

Worm burden, total peritoneal mf count, in vitro overnight mf secretion per worm, and qPCR data were analyzed using the Kruskal-Wallis Test followed by Dunn’s multiple comparison and using *t*-test. To determine significant differences in embryogram composition, a two-way ANOVA was conducted with Dunnett’s multiple comparisons test, and significance levels were determined based on comparisons with vehicle-treated animals. Expression of LAMP was analyzed using the *t*-test. All statistical analyses were determined using Prism 8 version 8.2.0.

### 2.8. Transmission Electron Microscopy of Female Brugia pahangi Worms

After overnight incubation, female worms were fixed with 2.5% glutaraldehyde and 2% paraformaldehyde buffered with 0.1M sodium cacodylate pH 7.3–7.4 as previously described [15]. After being submerged in fixative, the worms were cut into small pieces and kept for 2 h at room temperature before being stored at 4 °C. Following fixation, samples were washed with 0.1 M sodium cacodylate and post-fixed with 1% osmium tetroxide for 1 h. Following another wash with 0.1 M sodium cacodylate, samples were dehydrated in increasing ethanol concentrations (30–100%), washed two times with propylene oxide, and then infiltrated with a 1:1 Spurr:propylene oxide solution. Samples were then embedded in size 00 BEEM capsules using Spurr’s low-viscosity embedding kit (Electron Microscopy Sciences, Hatfield, PA, USA). Blocks were polymerized at 60 °C overnight. Ultrathin sections were collected on nickel formvar/carbon-coated 100 mesh grids or copper slot grids and contrasted with Uranyless (Electron Microscopy Sciences) and lead citrate. Sections were imaged on a Tecnai G2 Spirit TEM equipped with an AMT camera.

## 3. Results

### 3.1. In Vitro Screening of Autophagy-Inducing Compounds on Adult Female Brugia pahangi Worms

Nine known autophagy-inducing compounds (Appendix A) were used to treat adult females in vitro to study their effects on *Wolbachia* load, motility of worms, and their reproductive fitness (secretion of microfilariae in vitro and the composition of the developing embryonic stages within the female gonads). Based on the results of these assays (Appendix A), we selected the most effective compounds for in vivo testing (Appendix A). Some of the compounds were tested at different concentrations to find the optimal doses that would significantly affect worm fitness and the number of *Wolbachia*. Additionally, we used Doxycycline (12.5 µM) and DMSO (0.5–1%) as positive and negative controls, respectively.

The analyses showed that three of the nine compounds (Met-HCl, PI-103, and Clonidine) did not significantly reduce the total *Wolbachia* loads per worm, although the reduction levels were similar to what was observed with Doxycycline treatment (~33% decrease; Appendix A). Three other compounds (FK866, Imatinib, and Minoxidil) had no effect on *Wolbachia* loads in treated worms. However, 3 of the compounds were more effective than Doxycycline (Appendix A) and significantly reduced *Wolbachia* loads: 2-Thenoyltrifluoroacetone (TTFA) at 100 µM and 250 µM reduced *Wolbachia* levels by 46.4% and 53.6%, respectively; Rottlerin at 30 µM reduced it by 74.9%; and Niclosamide at 1 µM and 3 µM reduced the levels by 35% and 33.1%, respectively. Of all the in vitro compounds tested, 6 (TTFA, Imatinib, PI-103, Clonidine, Rottlerin, and Niclosamide) of the original 9 compounds significantly reduced the motility of worms and 4 (TTFA, PI-103, Rottlerin, and Niclosamide) of the 9 tested reduced the ability of treated female worms to secrete mf in overnight cultures (Appendix A).

We then selected Niclosamide and Rottlerin to test in vivo. We eliminated TTFA from in vivo testing because TTFA was effective only at a very high concentration (≥100 µM), which would be difficult to achieve in vivo. As Niclosamide was very potent in vitro, we also tested Niclosamide ethanolamine (NEN), a salt form of Niclosamide that has been demonstrated to uncouple mitochondrial oxidative phosphorylation [35,36] and which we found to be as potent as Niclosamide in vitro. We observed a significant reduction of *Wolbachia* loads in female worms, as well as reduced motility and ability to secrete mf (Appendix A).

Importantly, both Niclosamide and NEN were effective in reducing *Wolbachia* loads in a dose-dependent manner in adult female worms treated with 1 and 3 µM of Niclosamide and 0.3, 1, and 3 of µM of NEN (Appendix A, Figure 1A). Both compounds reduced *Wolbachia* loads in adult female worms by 45–69% and 60–91%, respectively, as compared to the DMSO control worms, with 1 µM and 3 µM Niclosamide and 0.1–3 µM of NEN inducing significant reductions (Figure 1A). Notably, both compounds at 3 µM reduced the motility of the female worms by 100% on day 6 in culture (Appendix A) and significantly reduced the secretion of microfilariae in vitro during the period between day 5 and 6 of treatment (Appendix A, Figure 1B). These compounds also affected the fitness of the developing embryonic stages within the gonads of the treated female worms (Figure 1C). In female worms treated with 1 µM Niclosamide or NEN, the number of deformed embryos was significantly elevated (*p* = 0.0286 for both). To verify that the effects of Niclosamide and NEN on the female worms were via their effect on autophagy and degradation steps that involve lysosomes, we tested the expression of the lysosome-associated membrane protein (LAMP) by Western blot. As shown in Figure 1D, both compounds induced the expression of LAMP by day 3 as compared to the DMSO control.

Moreover, Rottlerin at 30 µM, but not at 20 µM, significantly induced a 74.9% reduction in *Wolbachia* loads in adult female worms, a 68% reduction in their motility, and a 100% reduction in their ability to secrete microfilariae (Appendix A).

Based on the in vitro outcomes (Table 1), we selected Niclosamide, a known autophagy-inducing drug, for testing in vivo as it induced autophagy in the treated worms, caused a significant reduction of the *Wolbachia* loads in a dose-dependent way, and affected motility and secretion of microfilariae in treated female worms. As NEN was also effective in vitro and was shown to have better bioavailability than Niclosamide in other in vivo systems [35,36], we included it for comparison in our in vivo studies. Rottlerin was selected as a second drug for these proof-of-concept experiments.

### 3.2. In Vivo Pilot Testing of Niclosamide on the Fitness of Brugia pahangi Worms Using the Gerbil Infection Model

For the in vivo pilot study using the gerbil animal model of *Brugia* infection, we elected to test Niclosamide using two routes of drug administration (Appendix A, Experiment 1): (1) oral administration with 200 mg/kg and (2) IP administration with 10 mg/kg. We tested both types of compound administrations to compare and contrast the outcomes with the hope that the oral administration will also be effective and, therefore, can be repurposed for human use after testing in clinical trials. The IP route of administration is very common experimentally, as the *B. pahangi* worms reside within the peritoneal cavity. Both treatments were conducted for 10 days. Four to six days post-last treatment, the animals were sacrificed, and the recovered adult worms collected from the peritoneal cavity were analyzed. Treatments using both routes and doses did not result in any significant reduction in worm burden compared to the control group. However, qPCR analyses of DNA clearly showed a significant reduction of *Wolbachia* loads in the *B. pahangi* female worms recovered from the Niclosamide 200 mg/kg PO (*n* = 24; *p* < 0.0001) and the Niclosamide 10 mg/kg IP (*n* = 24; *p* < 0.05) as compared to the vehicle control group (*n* = 13) (Figure 2A). A portion of the recovered female worms from all groups (*n* = 35–51) were cultured overnight in vitro (Appendix A), and the number of secreted microfilariae per female worm was counted. Female worms from the Niclosamide 200 mg/kg PO group released significantly (*p* = 0.0278) fewer microfilariae as compared to the control group, while the number of microfilariae released from females from the Niclosamide 10 mg/kg IP group did not change significantly as compared to the vehicle control group (Figure 2B). Nonetheless, the embryogram analyses showed that female worms from both Niclosamide-treated groups had a higher proportion of deformed embryos (*p* = 0.0002) compared to the vehicle control group (Figure 2C,D). Additionally, analysis of LAMP expression by Western blot demonstrated a significant increase in its expression by 25% and 50% in Niclosamide 200 mg/kg PO and Niclosamide 10 mg/kg IP treated worms, respectively (Figure 2E).

### 3.3. Treatment of Infected Gerbils with 20 mg/kg Niclosamide and NEN Given IP Impacted the Fitness of Adult Female Worms

Based on the in vivo data with Niclosamide that supported the outcomes observed in vitro, the next in vivo experiment was improved by comparing Niclosamide with NEN-administrated IP and at a higher dosage of 20 mg/kg for both compounds. NEN, which is the ethanolamine salt of Niclosamide, has a similar safety profile as Niclosamide and better systemic exposure [35,36]. The dose increase was intended to enhance the efficacy of the treatments on parasite burden and fecundity. Hereafter and importantly, treatment intraperitoneally for 10 days with 20 mg/kg Niclosamide or NEN significantly reduced the total worm burden of both female and male worms by 68% and 50%, respectively (Figure 3A–C). The effects were more pronounced on male worms (Figure 3B, 82% and 60%, respectively) than on female worms (Figure 3A, 56% and 42%, respectively).

In addition, similarly to the outcomes we observed in vitro, both treatments significantly affected *Wolbachia* loads (*p* < 0.001, Figure 4A), reduced the secretion of microfilariae by 36% (Niclosamide; almost significantly, *p* = 0.065) and by 48% (NEN; *p* < 0.01, Figure 4B), as well as significantly reduced the total content of the embryonic stages within the gonads (*p* < 0.05 and *p* < 0.01, respectively) (Figure 4C). In addition, the number of deformed embryos was significantly increased 2–3-fold (*p* = 0.0187 for Niclosamide and *p* = 0.0169 for NEN) (Figure 4D). Notably, treatment also induced autophagy in the female worms recovered 4–6 days after the last treatment, as evidenced by Western blot analysis of the relative expression of LAMP vs. tubulin. The relative expression of LAMP was significantly induced in Niclosamide-treated female worms (25% increase, *p* < 0.01), but there was only a 15% increase in the NEN-treated worms (Figure 4E), as compared to females from the vehicle control group. These data suggest that the mechanism of action of both compounds on the filarial worms and *Wolbachia* is probably through the induction of autophagy.

Transmission electron microscopy (TEM) analyses of female worms recovered from treated gerbils indicated that the treatment also induced intracellular structural abnormalities in the embryonic stages within the female gonads. Compared to the vehicle groups (Figure 5A,B), worms recovered from gerbils treated with Niclosamide 20 mg/kg IP displayed deformation in the cells of the developing embryos, i.e., nuclei within the embryos were often ruptured or distorted (Figure 5C), and the cell membranes displayed robust losses of integrity, which is evident by the proximity of the deformed nuclei (Figure 5D). Worms recovered from NEN 20 mg/kg IP-treated gerbils also displayed abnormalities in the developing embryos (Figure 5E,F). Nuclear damage was frequently observed, and many nuclei displayed highly condensed, electron-dense chromatin that aggregated toward the middle of the nucleus.

### 3.4. Treatment of Infected Gerbils with Niclosamide and NEN In Vivo Using the Oral Route

As NEN was shown to have better bioavailability in other in vivo systems [35,36], we also decided to test whether oral administration, the optimal choice for drug administration, with 200 mg/kg NEN, would be more effective than the oral administration with 200 mg/kg Niclosamide in reducing survival and/or fitness of *Wolbachia* and the adult worms. Unfortunately, results showed that oral administration of both drugs had no effect on worm burden or *Wolbachia* loads nor on the fitness of female worms based on embryogram analyses. However, ultrastructural examination of the recovered worms revealed that treatment with both drugs caused damage to developing embryos as well as displayed a colocalization of *Wolbachia* and lysosomes in the hypodermis of the worms, which indicates an induction of autophagy-related process (Figure 6). Compared to the vehicle group (Figure 6A,B), worms recovered from gerbils treated with Niclosamide displayed complete destruction of nuclei within their embryos (Figure 6C). Additionally, *Wolbachia*-lysosome interactions in the hypodermis were prominent (Figure 6D), with instances where lysosomes were seen fusing with the bacteria (black asterisk). Worms recovered from NEN-treated gerbils also displayed abnormal embryos. Most notably, within the embryos, nuclear damage in the form of ruptured membranes was observed, and many nuclei contained abnormal chromatin (Figure 6E,F).

Next, we tested a new treatment regimen that could increase the bioavailability of the drugs: treatment twice a day (BID) orally with Niclosamide 100 mg/kg or NEN 100 mg/kg for 10 days. Notably, this new regimen was more effective in decreasing the survival of the adult worms, but only in the Niclosamide BID-treated group, whereas this was not seen in the NEN BID-treated group (Appendix A, Experiment 4). In the Niclosamide group, there was a 72% reduction in total worm burden (82% reduction in females and 50% reduction in males). As only eight female worms were recovered from the Niclosamide-treated animals in total, we could not analyze the other parameters of worm fitness or *Wolbachia* loads, but we were able to analyze the fixed female worms by TEM.

Markedly, even if there were no effects on worm burden or *Wolbachia* loads in the worms recovered from NEN 100 mg/kg BID-treated gerbils, the worms displayed significant ultrastructural damage. In comparison to structures in worms recovered from vehicle-treated gerbils (Figure 7A,B), NEN 100 mg/kg PO BID-treated worms had many abnormally shaped nuclei with apparent ruptures (Figure 7C). Additionally, cell membranes were also affected, which is evident by the proximity of many deformed nuclei and the lack of integral cell membranes (Figure 7D). Importantly, although there were only minor structural deformities in the Niclosamide 100 mg/kg PO BID worms, ultrastructural analyses confirmed that treatment increased the number of hypodermal lysosomes that often appeared in large clusters and of varying maturities (Figure 7E). Moreover, the number of hypodermal lysosomes was significantly increased (Figure 7F).

### 3.5. Treatment of Infected Gerbils with Rottlerin In Vivo Using the Oral Route

Rottlerin showed a robust reduction of *Wolbachia* load, motility, and secretion of microfilariae when used at 30 µM in vitro (Appendix A). We, therefore, also explored the potency of Rottlerin in vivo and its potential use as a repurposed candidate for this proof-of-concept study. As the compound was shown to be effective in other systems when administered orally [37,38], we treated infected gerbils with Rottlerin 50 mg/kg PO, BID, for 10 days. Although the treatment had no effect on worm burden, there was a significant decrease in *Wolbachia* loads (Figure 8A) in the worms recovered from treated gerbils, as we observed in vitro (Appendix A). Although treatment using this regimen had no effect on the total gonad content (Figure 8B) or the secretion of microfilariae per worm (Figure 8C), it appeared to affect their embryogenesis and induce almost four times more deformed embryos in the gonads of the treated worms (Figure 8D). Importantly, based on the LAMP assay, autophagy was significantly induced in the recovered female worms (Figure 8E), but it was not sufficient in this single experiment to result in decreased worm burden.

Ultrastructural analyses revealed significant abnormalities in the gonads of the female worms recovered from gerbils treated orally with Rottlerin 50 mg/kg BID (Figure 9B,C) in comparison to the vehicle group (Figure 9A). Most notably, treated worms (Figure 9B,C) displayed embryos with numerous debris-filled autophagosomes (white arrowheads). Some embryos (Figure 9C) also contained large, mature lysosomes. In addition to structural abnormalities, Rottlerin-treated worms also showed a significant increase in hypodermal lysosomes (Figure 10B) compared to vehicle treated worms (Figure 10A,C).

## 4. Discussion

Targeting *Wolbachia*, the endosymbiotic bacteria of filarial parasites, is an effective screening platform for novel therapeutics against the adult stages of filarial worms. Since *Wolbachia* are Gram-negative bacteria, specific antibiotics have been used to eliminate the bacteria from their filarial host, leading to the eventual killing of the adult worms. Clinical trials with doxycycline have shown dramatic depletion (more than 99%) of *Wolbachia* from filarial nematodes (6-week treatment, 200 mg/day) that resulted in significant reduction of microfilaridermia and killing of *O. volvulus* adult worms [39] through the activation of apoptosis [40]. Importantly, treatment with such antibiotics is beneficial when people are co-infected with *L. loa*, a human filarial parasite that does not harbor an endosymbiont [39] but one, which, in high numbers, causes severe adverse events when individuals are treated with the drug ivermectin [41]. Trials in onchocerciasis and loiasis co-endemic regions in Africa have confirmed that antibiotics can be delivered safely without inducing severe pathology associated with the treatment of co-infected people with ivermectin. Despite the discovery that doxycycline, minocycline, and rifampicin showed important macrofilaricidal effects in clinical trials, they have the disadvantage of requiring administration for 4 or more weeks and cannot be given to children below the age of 9 and pregnant women [42]. The duration of the treatment regimen is a particularly important limitation as it negates its use for a widespread scale-up or MDA in resource-poor settings, such as those found in Sub-Saharan Africa. This has motivated the research community to identify alternative antibiotic-like drugs that can be effective over short treatment courses, such as *Flubentylosin*, a derivative of Tylosin, a veterinary antibiotic that targets *Wolbachia* and that is currently being tested in clinical trials [43,44]. Based on preclinical pharmacokinetic and pharmacodynamic modeling, Flubentylosin is expected to be effective at a dose of 400 mg taken daily for 7 or 14 days. 

Recently, alternative approaches are being investigated, shifting the focus from bacteriostatic drugs that kill the bacteria to the processes that are essential for mutualistic interactions between the filarial parasite and *Wolbachia*. For example, we recently showed that glycolysis plays an essential role in the symbiosis between *Wolbachia* and *Brugia* worms and that suppression of glycolysis significantly reduced pyruvate (the end-product of glycolysis) in *Brugia* worms. We hypothesized that the pyruvate delivered from the filarial host is needed for *Wolbachia* and used for energy and purine/pyrimidine pathways [30,45]. These bacterial pathways are predicted to be essential for worm fitness. Treatment with compounds that inhibit glycolytic enzymes caused detrimental effects on both *Wolbachia* and its *Brugia* host, suggesting that alternative sets of compounds can target host pathways to disrupt symbiosis in adult worms [30,45].

Another attractive alternative approach, which we have explored in this study, is the use of repurposed drugs that can initiate the intracellular innate defense mechanisms present in the filarial worms known to help eukaryotic cells recognize and eliminate intracellular bacteria [27]. Eukaryotic cells use autophagy to find and degrade intracellular invaders, including parasites, bacteria, and viruses. Prior research in various systems has shown that in symbiotic associations between *Wolbachia* and eukaryotic cells, autophagy plays an important role in controlling the intracellular population of the bacteria [27]. Importantly, in *Brugia* worms, the induction or suppression of autophagy changed the size of *Wolbachia* populations. Moreover, induction of autophagy also increased the number of lysosomes in *Brugia*, which directly fused with vacuoles containing *Wolbachia*. These experiments confirmed the potential for a direct approach to inducing the degradation of the endosymbiont in the filarial parasites.

In this study, we tested whether compounds known to induce autophagy in eukaryotic cells can also cause the filarial worms to target their own *Wolbachia* symbiont in vitro and in vivo and, consequently, affect worm fitness, thus strengthening the potential of such a novel strategy. We posit that autophagy-mediated degradation of *Wolbachia* in filarial worms will significantly reduce the bacterial load and/or its molecular components that are essential for the mutualistic interplay between the parasite and its *Wolbachia* symbiont. This could promote new development programs to repurpose autophagy-inducing drugs for new indications that effectively target infections that cause lymphatic filariasis and other filarial diseases.

In our proof-of-concept study, we used a standard workflow where we first screened a small library of known autophagy inducers on female adult worms in vitro and then validated the effects of a few selected drugs on the worms in an animal model. In the initial in vitro tests, we used nine compounds. Three of them (Niclosamide, NEN, and Rottlerin) showed more than a 35% reduction in *Wolbachia* loads in adult female worms after 6 days of treatment (Appendix A) as compared to Doxycycline, an antibiotic known to reduce filarial *Wolbachia* load in vitro and in vivo. The three compounds also significantly reduced motility and completely blocked the secretion of microfilariae by the treated female worms (Appendix A).

We then used the gerbil animal model for filarial infection [46,47], widely used for in vivo screening of compounds that are candidates for anti-filarial treatment [27,48,49,50] to test Niclosamide, NEN, and Rottlerin for their effects on the adult worms in vivo. We used a battery of assays commonly used in both in vitro and in vivo experiments that include worm burden, *Wolbachia* loads, worm motility, and female fecundity (microfilarial secretion and embryograms). Moreover, we used protein analyses to confirm that the drugs induced autophagy (i.e., LAMP assay), which was also validated by electron microscopy (induction of lysosomes).

Our initial experiments with Niclosamide (200 mg/kg PO and 10 mg/kg IP) clearly showed that treatment can significantly reduce *Wolbachia* loads and, to some extent, the fecundity of the female worms recovered from the treated animals without affecting their survival (Figure 2). To increase the effectiveness of Niclosamide, we tested the intraperitoneal route with 20 mg/kg Niclosamide and compared it to treatment with NEN using this higher dose (Figure 3, Figure 4 and Figure 5). NEN is a Niclosamide ethanolamine salt shown to have better bioavailability in other in vivo systems [35,36]. We confirmed that NEN has a similar effect as Niclosamide in vitro on the reduction of *Wolbachia* loads and affects motility and the secretion of microfilariae from treated female worms (Appendix A, and Figure 1). Importantly, intraperitoneal treatment for 10 days with the higher dose of 20 mg/kg of Niclosamide or NEN significantly reduced the total *Wolbachia* load and the total gonad content per worm as well as significantly increased the percentage of deformed embryos within the gonads. Moreover, both treatments significantly reduced the total female and male worm burden, while treatment with NEN IP also reduced microfilariae secretion more effectively and significantly (Figure 3 and Figure 4). The effects on fecundity were confirmed by TEM analysis that showed clear ultrastructural deformities of the nuclei and cell membranes in the embryonic stages of the parasite (Figure 5). In addition, female worms recovered from gerbils treated with both drugs expressed LAMP, where it was significantly induced in Niclosamide-treated worms and almost significantly (15% increase; *p* = 0.065) in NEN-treated worms (Figure 4).

Oral administration with Niclosamide (200 mg/kg) or NEN (200 mg/kg), however, did not have the same effects as the intraperitoneal treatment with Niclosamide or NEN using 20 mg/kg IP. Notwithstanding, ultrastructural examination of the surviving female worms showed damaged nuclei within an embryo displaying large ruptures, abnormal chromatin leaking from the nuclear membrane and nuclei within embryos displaying large, abnormal chromatin aggregates and ruptured nuclear membranes, as observed with the intraperitoneal treatment (Figure 6). To potentially increase bioavailability when using the oral route, we next examined whether two oral treatments per day (BID) with 100 mg/kg Niclosamide or NEN for 10 days would improve effectiveness of the treatment on worm survival and/or fitness. Remarkably, BID dosing with Niclosamide showed a significant reduction in worm burden (72% reduction compared to the control group). However, this was not observed in the NEN BID-treated group. The recovered female worms from gerbils treated with Niclosamide BID had an increased number of hypodermal lysosomes (Figure 7). Notably, ultrastructural analyses of the recovered female worms from the NEN group (Figure 7) had similar abnormalities as the Niclosamide-treated worms (Figure 6) in embryos within the gonads: abnormally shaped nuclei with an apparent rupture, ruptured nuclei, and lack of cell membranes separating the nuclei (Figure 7). However, none of these worms had an increased number of hypodermal lysosomes, which might explain the lack of NEN BID effect on worm burden.

Oral administration with our second autophagy-inducing drug, Rottlerin (50 mg/kg BID for 10 days), also had no effect on worm burden, total gonad content of microfilariae per worm, or the secretion of microfilariae. However, there was a significant decrease in *Wolbachia* load in the worms recovered from treated gerbils and a 4-fold increase in the number of deformed embryos in the gonads (Figure 8, Figure 9 and Figure 10). Importantly, based on the LAMP assay, autophagy was significantly induced in the recovered female worms, but in this single experiment, it was not sufficient to result in decreased worm burden.

The in vivo data, taken together, corroborates the findings obtained from the in vitro studies, demonstrating good in vitro to in vivo translation. In vitro, Niclosamide, NEN, and Rottlerin also significantly affected female worm fecundity, as evidenced by the decreased microfilariae secretions, elevated numbers of deformed embryos, reduced *Wolbachia* levels, and the induction of lysosome-associated membrane proteins. Results of the extensive in vitro assays conducted prior to moving forward with the in vivo studies helped to guide the prioritization and selection of repurposed drugs for use in the animal studies and showed, at the end of the in vivo investigations, that the effects of the drugs on worms in culture were similar to those seen in the gerbil host.

Taking advantage of the fact that *Wolbachia* populations in different *Wolbachia*-host biological systems are under direct and continuous control by autophagy, we have shown that induction of autophagy can be a novel way to turn the filarial parasites against their endosymbionts. We found that autophagy-inducing compounds—Niclosamide, NEN, and Rottlerin, which are three drugs used safely in human studies [51,52,53]—can reduce *Wolbachia* in filarial worms and affect female worm fitness in vitro and in vivo. Future studies to determine optimal treatment doses, regimens, and time needed to conduct post-treatment analyses are warranted as this would allow more in-depth analyses of long-term effects that treatments with such repurposed autophagy-inducing compounds can have on filarial worm burden and fitness.

## Figures and Tables

**Figure 1 pharmaceutics-16-00256-f001:**
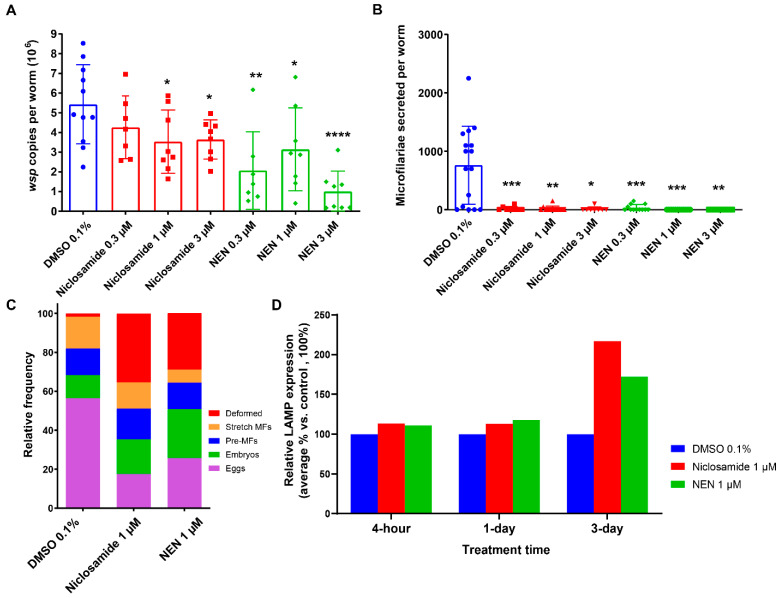
Treatment with Niclosamide and NEN in vitro reduced *Wolbachia* loads and worm fecundity and led to upregulation of lysosomal marker. (**A**) *Wolbachia* loads in treated and non-treated female worms were quantified using qPCR of the single copy *wsp* gene. (**B**) The number of microfilariae secreted from individual adult female worms was quantified over days 5–6 in culture. (**C**) Embryogram analyses using phase contrast microscopy were performed on individual female worms 6 days post-treatment, and the numbers of the various embryonic developmental stages (eggs, embryos, pre-microfilariae, stretched microfilariae, deformed embryos) in at least 200 events were determined. (**D**) Analysis of LAMP expression of treated worms (*n* = 2, per sample) by Western blot. LAMP values are normalized to tubulin expression. *p*-value < 0.05 (*), *p*-value < 0.01 (**) and *p*-value < 0.001 (***), *p*-value < 0.0001 (****).

**Figure 2 pharmaceutics-16-00256-f002:**
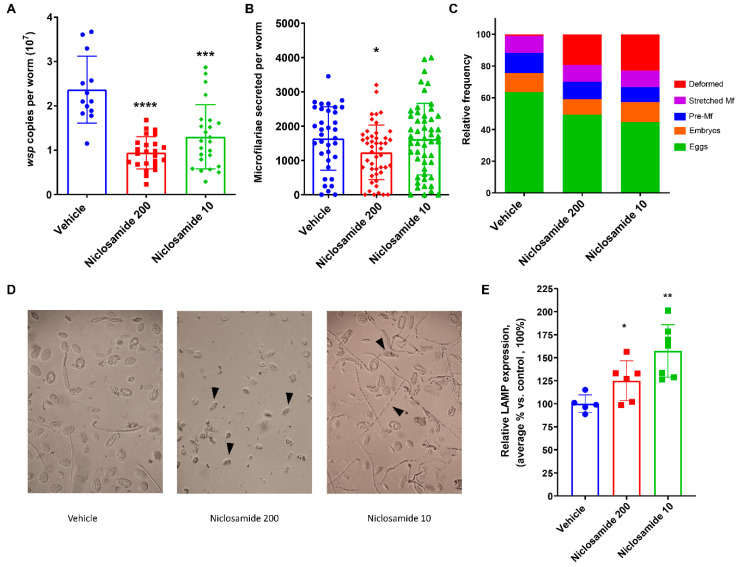
In vivo treatment of infected gerbils with Niclosamide 200 mg/kg PO and 10 mg/kg IP reduced *Wolbachia* loads and induced deformity in worm progeny after a 10-day treatment regimen. (**A**) *Wolbachia* loads in female worms recovered from treated gerbils were quantified using qPCR of the single copy *wsp* gene. (**B**) The number of microfilariae secreted from individual adult female worms was quantified over days 5–6 in culture. (**C**) Embryogram analyses using phase contrast microscopy were performed on individual female worms 6 days post-treatment, and numbers for various embryonic developmental stages (eggs, embryos, pre-microfilariae, stretched microfilariae, deformed embryos) in at least 200 events were determined. (**D**) Representative phase contrast micrographs from the embryogram analyses. Black arrowheads demonstrate deformed progeny in treated worms. (**E**) Analysis of LAMP expression of treated worms by Western blot. LAMP expression is normalized to tubulin expression. *p*-value < 0.05 (*), *p*-value < 0.01 (**) and *p*-value < 0.001 (***), *p*-value < 0.0001 (****).

**Figure 3 pharmaceutics-16-00256-f003:**
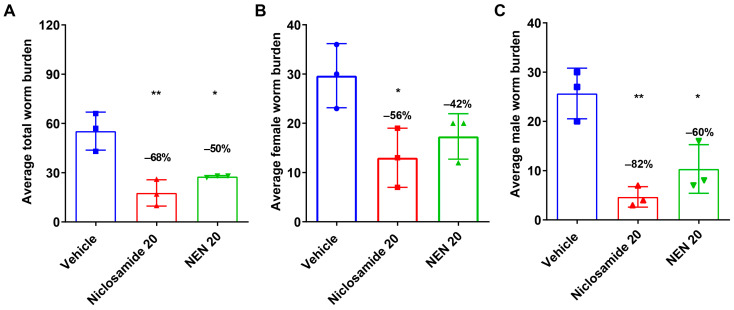
Treatment of infected gerbils with 20 mg/kg Niclosamide and NEN using the intraperitoneal route reduced worm burden after a 10-day treatment regimen. (**A**) Mean number of total worm burden per animal; (**B**) Mean number of female worm burden per animal; and (**C**) Mean number of male worm burden per animal. Necropsies were performed 4–6 days after final treatment, and worms were recovered from the peritoneum. *p*-value < 0.05 (*), *p*-value < 0.01 (**).

**Figure 4 pharmaceutics-16-00256-f004:**
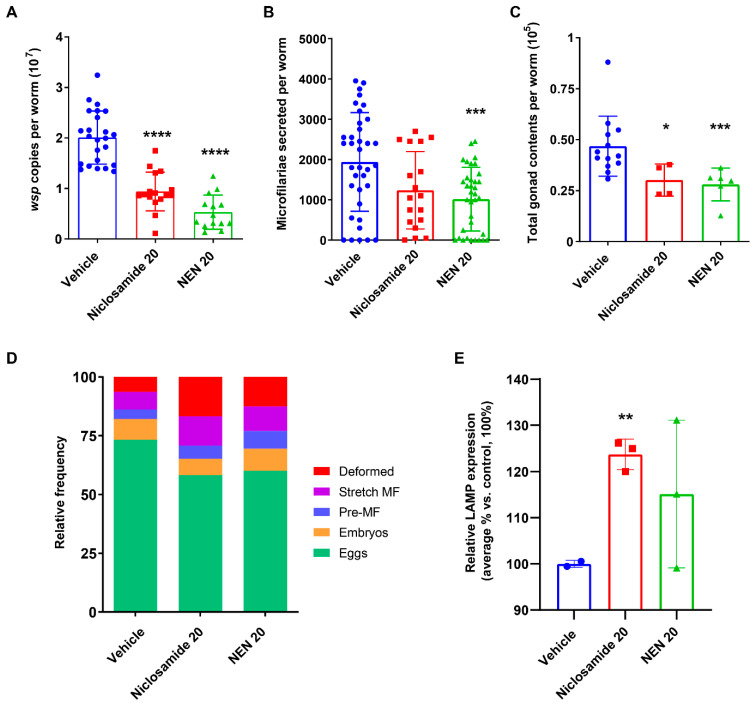
In vivo treatment of infected gerbils with Niclosamide 20 mg/kg and NEN 20 mg/kg IP affected *Wolbachia* loads, worm fecundity, and upregulated markers of autophagy after a 10-day treatment regimen. (**A**) *Wolbachia* loads in female worms recovered from treated gerbils were quantified using qPCR of the single copy *wsp* gene. (**B**) The number of microfilariae secreted from individual adult female worms was quantified over days 5–6 in culture. (**C**) Quantification of total gonad content in female worms. (**D**) Embryogram analyses using phase contrast microscopy were performed on individual female worms 6 days post-treatment, and the numbers of the various embryonic developmental stages (eggs, embryos, pre-microfilariae, stretched microfilariae, and deformed embryos) in at least 200 events were determined. (**E**) Analysis of LAMP expression of treated worms by Western blot. LAMP expression is normalized to tubulin expression. *p*-value < 0.05 (*), *p*-value < 0.01 (**) and *p*-value < 0.001 (***), *p*-value < 0.0001 (****).

**Figure 5 pharmaceutics-16-00256-f005:**
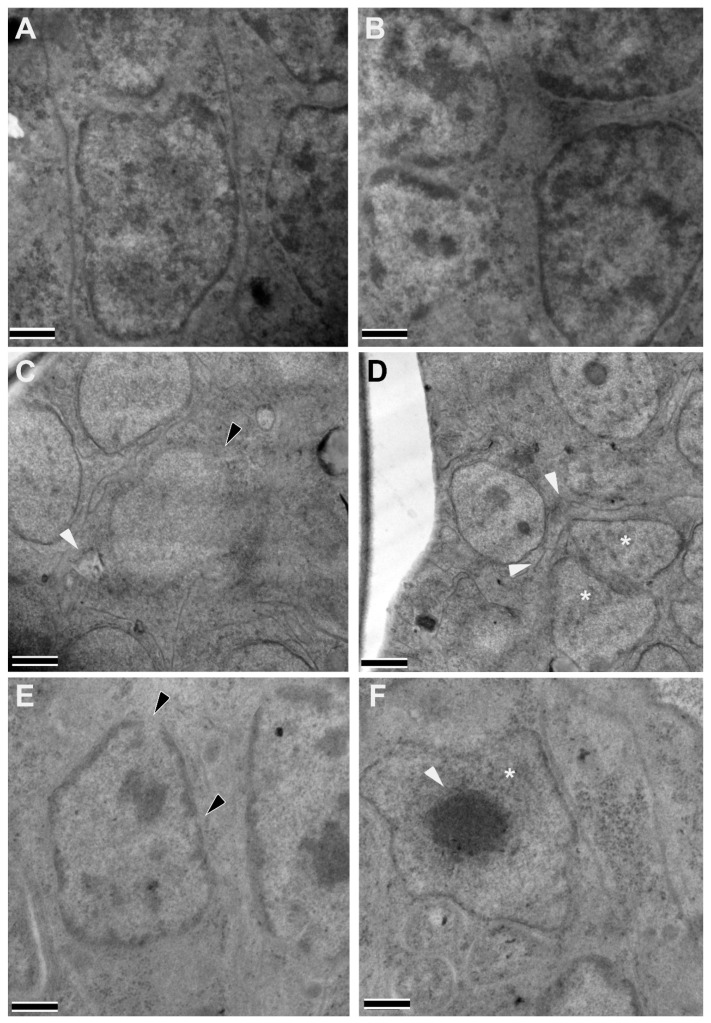
Intracellular structural alterations induced in adult female worms by the treatment of infected gerbils with 20 mg/kg Niclosamide and NEN IP. Female worms recovered from gerbils treated with Vehicle ((**A**,**B**), normal nuclei, and cell membranes), 20 mg/kg Niclosamide IP (**C**,**D**), or 20 mg/kg NEN IP (**E**,**F**) were fixed and evaluated by transmission electron microscopy. A large rupture in the cell membrane (white arrowhead) and nuclear membrane (black arrowhead) within an embryo in Niclosamide-treated worms (**C**), and two deformed nuclei (asterisks) in contact with each other and large ruptures in the cell membranes (white arrowheads) within an embryo (**D**). In NEN-treated worms, a damaged nucleus (**E**) with apparent ruptures (black arrowheads) and an abnormal nucleus (asterisk) containing highly condensed and abnormal chromatin (white arrowheads) (**F**) can be seen. (Scale bars = 2 µm).

**Figure 6 pharmaceutics-16-00256-f006:**
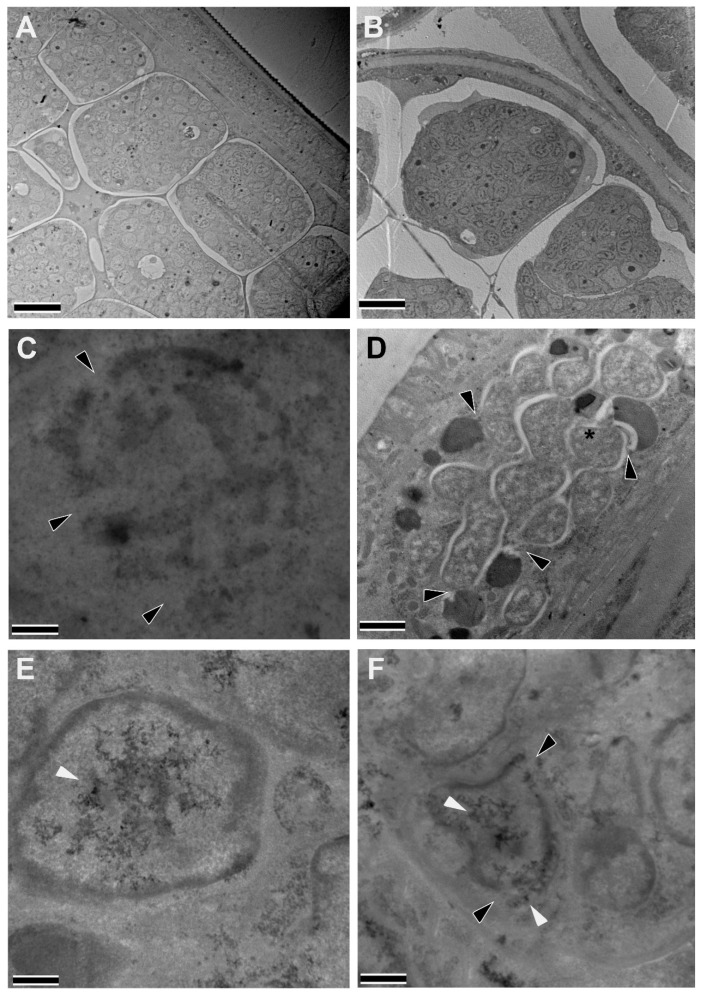
Intracellular structural alterations induced in adult female worms recovered from infected gerbils treated with 200 mg/kg Niclosamide or NEN PO. Female worms recovered from gerbils treated with Vehicle (**A**,**B**), Niclosamide (**C**,**D**), or NEN (**E**,**F**) were fixed and evaluated by transmission electron microscopy. (**A**,**B**) In vehicle-treated gerbils, the female worms had a normal hypodermis and gonad and developing embryos with healthy nuclei. In Niclosamide-treated gerbils, note a destroyed nucleus (**C**) with large ruptures in the nuclear membrane (black arrowheads) and the numerous *Wolbachia*-lysosome interactions (black arrowheads) in the hypodermis of a recovered treated worm (**D**) with one bacterium (black asterisk) fusing with the lysosome. In NEN-treated gerbils, the gonad of the female worms (**E**) had a nucleus containing abnormal chromatin (white arrowhead) within an embryo as well as (**F**) a damaged nucleus with large ruptures (black arrowheads) and abnormal chromatin (white arrowheads). (Scale bars—**A**,**B** = 10 µm; **C**–**F** = 2 µm).

**Figure 7 pharmaceutics-16-00256-f007:**
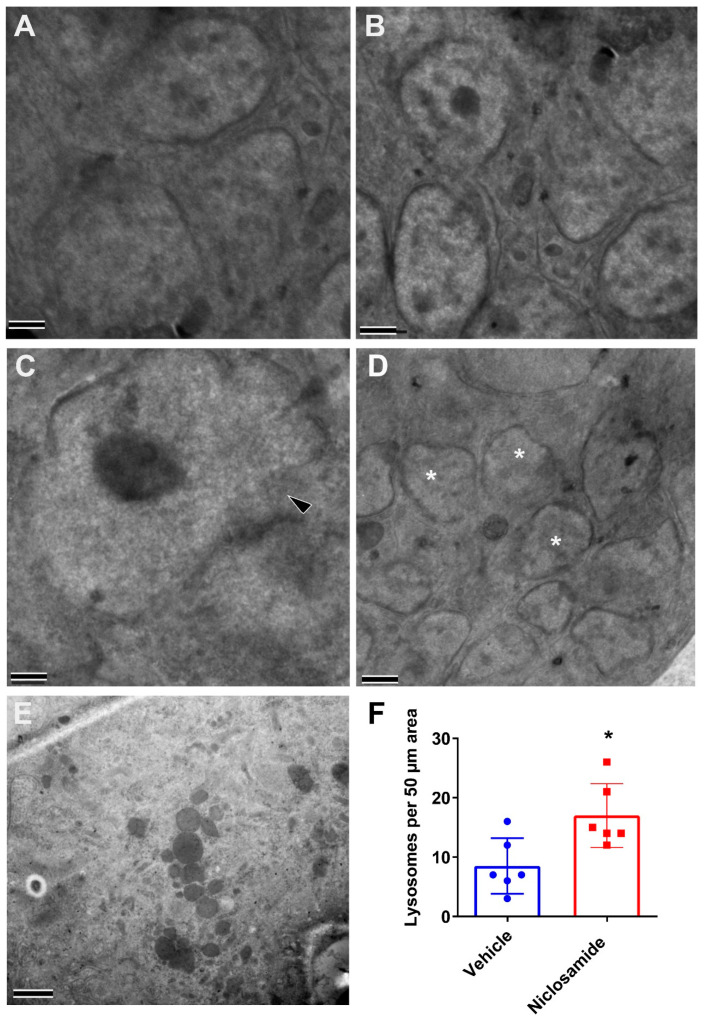
Intracellular structural alterations induced in adult female worms from infected gerbils treated with 100 mg/kg Niclosamide and NEN PO BID. Female worms recovered from gerbils treated with Vehicle (**A**,**B**), NEN (**C**,**D**), or Niclosamide (**E**) were fixed and evaluated by transmission electron microscopy. In comparison to normal nuclei within a developing embryo in the vehicle group (**A**,**B**), the NEN-treated group had abnormal nuclei (**C**) with a large rupture (black arrowhead), as well as several deformed nuclei (asterisks) with apparent ruptures in proximity to each other within an embryo (**D**). In Niclosamide-treated worms, there was an increased number of hypodermal lysosomes (**E**). The hypodermal lysosomes were quantified (**F**) by imaging 6 randomly selected 50 µm regions of the hypodermis from each treatment condition (*, *p* < 0.05). (Scale bars = 2 µm).

**Figure 8 pharmaceutics-16-00256-f008:**
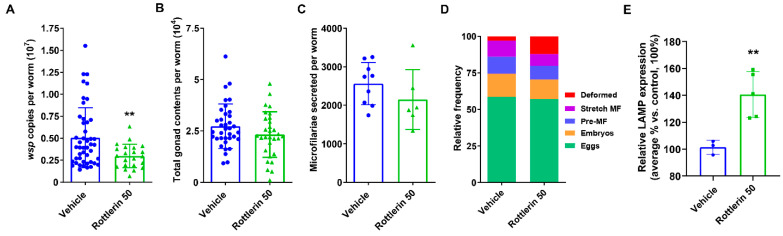
Treatment of infected gerbils with 50 mg/kg Rottlerin PO and BID affected *Wolbachia* loads but not fitness of recovered female worms. (**A**) *Wolbachia* loads in female worms recovered from treated gerbils were quantified using qPCR of the single copy *wsp* gene. (**B**) Quantification of the total gonad content in female worms. (**C**) The number of microfilariae secreted from individual adult female worms was quantified over days 5–6 in culture. (**D**) Embryogram analyses 6 days post-treatment; the numbers of the various embryonic developmental stages (eggs, embryos, pre-microfilariae, stretched microfilariae, and deformed embryos) were counted in at least 200 total events. (**E**) Relative LAMP expression in treated worms vs. vehicle by Western blot. *p*-value < 0.01 (**).

**Figure 9 pharmaceutics-16-00256-f009:**
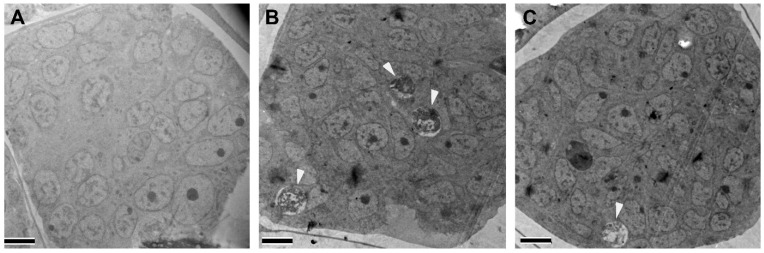
Intracellular structural alterations are induced in adult female worms recovered from infected gerbils treated with 50 mg/kg Rottlerin PO and BID. Female worms recovered from gerbils treated with vehicle (**A**) or Rottlerin (**B**,**C**) were fixed and evaluated by transmission electron microscopy. In comparison to a normal embryo with healthy nuclei (**A**), Rottlerin-treated worms (**B**) had numerous autophagosomes (white arrowheads) within an embryo as well as a large lysosome and a debris-filled autophagosome (white arrowhead) within an embryo (**C**). (Scale bars = 5 µm).

**Figure 10 pharmaceutics-16-00256-f010:**
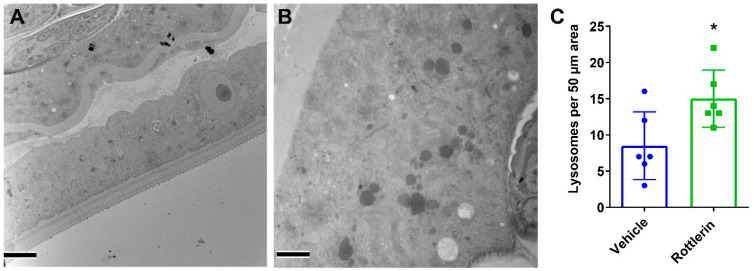
Intracellular structural alterations induced in adult female worms by the treatment of infected gerbils with 50 mg/kg Rottlerin PO and BID. Representative images of regions of hypodermis from vehicle (**A**) and Rottlerin (**B**) treated worms show a significant increase of hypodermal lysosomes in the Rottlerin-treated worms. (**C**) Lysosomes were quantified by imaging six randomly selected 50 µm regions of the hypodermis from each treatment condition (*, *p* < 0.05). (Scale bars = 5 µm).

**Table 1 pharmaceutics-16-00256-t001:** Highlights of results from in vitro experiment. For details of statistical data, see Appendix A. *, *p* < 0.05; **, *p* < 0.01; ***, *p* < 0.001; ****, *p* < 0.0001.

Compound	Concentration	% Change *wsp* Copies	% Change of Motility	% Change in mf Secretion
**Niclosamide**	0.3 µM	−12%	−30.2% **	−95% ****
	1 µM	−35% *	−49.7% ****	−98% ***
	3 µM	−33.1% **	−100% ****	−98% **
**NEN**	0.3 µM	−39.6% *	−23.1% **	−93% ****
	1 µM	−42.2% *	−53% ****	−100% ***
	3 µM	−81.6% ****	−100% ****	−100% **
**Rottlerin**	30 µM	−74.9% ***	−68.8% ****	−100% *

## Data Availability

The original contributions presented in the study are included in the article/Appendix A, further inquiries can be directed to the corresponding authors.

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
