# Peer review of "Repurposed Drugs That Activate Autophagy in Filarial Worms Act as Effective Macrofilaricides"

_pharmaceutics, 2024, doi:10.3390/pharmaceutics16020256_

Round 1

Reviewer 1 Report

Comments and Suggestions for Authors

Repurposed drugs that activate autophagy in filarial worms act as effective macrofilaricides

Manuscript ID: pharmaceutics-2810398 By: Denis Voronin, Nancy Tricoche, Ricardo Peguero, Anna Maria Kaminska, Elodie Ghedin, Judy A Sakanari, Sara Lustigman

This manuscript presents research regarding the utilization of repurposed drugs with the intentions of treating Brugia Pahangi infections. Through their in vitro and in vivo studies, the authors demonstrate the potential for targeting novel alternative therapeutics approaches for onchocerciasis and lymphatic filariasis by using Wolbachia as a chemotherapeutic target. The study is presented in an organized manner with sound research design that is backed by peer-reviewed research and logic. For these For these reasons, we recommend the manuscript be accepted with minor revisions.

Major comments:

1. In the abstract, the authors often state that Niclosamide and Rottlerin ‘affected’ the fecundity and embryogenesis in recovered female worms and niclosamide and NEN ‘impacted’ adult worm survival. It would be nice for the authors to concisely elaborate the specific effects of the drugs on the investigated parameters.
2. Please present the data for the in vitro studies in the main manuscript. Considering the in vitro data was employed to select the drugs for the in vivo studies, it might be worthwhile to show this data in the main manuscript for the readers to follow along with the repurposing of the drugs.
3. In the in vitro studies, the authors utilized various concentrations of the compounds. How were these concentrations chosen? Were the inhibitory effects of the compounds on Wolbachia in adult female Brugia pahangi worms analyzed with the variations in the concentrations in mind?
4. Throughout the manuscript, the authors employ the oral and intraperitoneal modes of administration. The way these studies are presented are rather unorganized and the logical design in the order of these studies can be confusing. Please address this issue.

Minor comments:

1. If possible, make connections between the in vitro and in vivo data.
2. The in vivo efficacy of the compounds (niclosamide, NEN, and rottlerin) often do not exhibit dosedependence. What do you think the reason for this is?

The overall concept of the manuscript could be of interest to the readership of Pharmaceutics with regards to the repurposing of drugs in developing novel alternative treatment methods for onchocerciasis and lymphatic filariasis. Furthermore, this study demonstrates the possibility of chemically targeting Wolbachia to decrease the fecundity, embryogenesis, and adult worm survival of filarial worms. Based on these notions, we recommend the acceptance of this research manuscript to Pharmaceutics.

Author Response

  1. In the abstract, the authors often state that Niclosamide and Rottlerin ‘affected’ the fecundity and embryogenesis in recovered female worms and niclosamide and NEN ‘impacted’ adult worm survival. It would be nice for the authors to concisely elaborate the specific effects of the drugs on the investigated parameters.

Response: Thank you for your comment. We have now changed the text in the abstract. Lines: 35-44.

  1. Please present the data for the in vitro studies in the main manuscript. Considering the in vitro data was employed to select the drugs for the in vivo studies, it might be worthwhile to show this data in the main manuscript for the readers to follow along with the repurposing of the drugs.

Response: Thank you for this suggestion. We have now added Table 1 (Lines: 265, 302) that summarizes the key results of the in vitro data to the Results section. Readers can also view more detailed information on these studies in the Supplemental Tables S1-S3.

  1. In the in vitro studies, the authors utilized various concentrations of the compounds. How were these concentrations chosen? Were the inhibitory effects of the compounds on Wolbachia in adult female Brugia pahangi worms analyzed with the variations in the concentrations in mind?

Response: This is the first study to assess the effects of autophagy-inducing compounds on filarial worms. Some of the starting concentrations were based on those from the literature for in vitro treatment Brugia worms and using these drugs on other biological systems. We then used various concentrations to optimize the studies both in vitro and in vivo. We have clarified this point in the results section, lines 242-244, 352-353, 449-452.

  1. Throughout the manuscript, the authors employ the oral and intraperitoneal modes of administration. The way these studies are presented are rather unorganized and the logical design in the order of these studies can be confusing. Please address this issue.

Response: We have now added explanations for each of the experimental steps in the manuscript (at the begin of each result sections).  Lines: 309-313, 348-353, 365-366, 418-423, 449-452, 482-488.  We believe it is clearer now.

Minor comments:

  1. If possible, make connections between the in vitro and in vivo data.

Response: Thank you for your suggestion. We added the statement in the Discussion Lines: 647-655.

  1. The in vivo efficacy of the compounds (niclosamide, NEN, and rottlerin) often do not exhibit dose dependence. What do you think the reason for this is?

Response: Since the in vivo studies were meant to demonstrate proof-of-concept, we did not investigate dose dependency per se but rather we investigated the effects of the drugs in animals treated by oral (PO) or intraperitoneal (IP) administration as well as by varying the dosage and number of times the drugs were administered (once a day vs BID).  Each of the studies was based on the findings from the previous experiments, e.g. IP to PO dosing, once a day to BID. Further in vivo studies would include a wider range of concentrations, shorter dosing treatments and longer post-treatment periods.

Reviewer 2 Report

Comments and Suggestions for Authors

This manuscript is dedicated for search of new possibilities for the treatment of neglected tropical diseases Onchocerciasis and lymphatic filariasis. Authors suggested a successful approach to target Wolbachia, obligatory endosymbiotic bacteria on which filarial worms are dependent for their survival and reproduction within the human host.

In this work authors applied a drug repurposing method and selected 9 known compounds for inducing of autophagy, which could disturb various pathways involved in the interdependency between Wolbachia and filarial worms.

Authors demonstrated that several such compounds, including Niclosamide and Rottlerin signifcantly reduced the levels of Wolbachia and impacted adult worm survival. These repurposed drugs can provide a new avenue for the clearance of adult worms in filarial infections.

This paper is well-suitable for the Special Issue „Emerging Pharmaceutical Therapeutics for Neglected Tropical Diseases“. The manuscript is well-thought, original, interesting and very detailed. Thus it only needs a few minor improvements and small corrections.

1. Probably not so many readers of paper will know such drugs as PI-183, Rottlerin or FK866, so, it can be recommended to insert Figure with chemical structures of all 9 compounds, used in this work.

2. In Materials and Methods please also add an additional information about purity (%) and source (Company) for each used compound.

3. References must be improved and corrected according to the requirements for MDPI Journals.
3.1. At the end of line 736 please add 189 (2022, 15, 189.).
3.2. Please provide in all your references journals abbreviations with points (for example, line 745, in Ref. 38: correct as „Front. Vet. Sci.“, similar in all others.
3.3. Please use only generally accepted journal abbreviations (ISO 4 Standard) (re-check in Google) and insert DOI indexes for all citted papers: for Ref. 1: doi: 10.1016/S0140-6736(10)60586-7 and similar for all others, which haven‘t.

After all listed minor corrections will be done, the current manuscript can be accepted for publication.

Author Response

1. Probably not so many readers of paper will know such drugs as PI-183, Rottlerin or FK866, so, it can be recommended to insert Figure with chemical structures of all 9 compounds, used in this work.

Response: Thank you for your suggestion. We have now included this information (purity and structure) in the supplement Table S1.

2. In Materials and Methods please also add an additional information about purity (%) and source (Company) for each used compound.

Response: We added this information to Table S1.

3. References must be improved and corrected according to the requirements for MDPI Journals.
3.1. At the end of line 736 please add 189 (2022, 15, 189.).
3.2. Please provide in all your references journals abbreviations with points (for example, line 745, in Ref. 38: correct as „Front. Vet. Sci.“, similar in all others.
3.3. Please use only generally accepted journal abbreviations (ISO 4 Standard) (re-check in Google) and insert DOI indexes for all citted papers: for Ref. 1: doi: 10.1016/S0140-6736(10)60586-7 and similar for all others, which haven‘t.

Response: We have corrected and formatted references using EndNote according to the journal’s requirements. We manually edited the references according to Reviewer recommendations. We added the DOIs where it was possible.

Reviewer 3 Report

Comments and Suggestions for Authors

This is an original and quite interesting comprehensive paper addressed to the search for additional drugs that improve treatment of filiarial infections mainly caused by Brugia pahangi and Onchocerca volvulus, based on microscopic (optical and electronic), biochemical and molecular studies as well as experimental  infections in gerbils.

The results clearly demonstrate and evidenced that both Niclosamide and Rottlerin (a natural product from Kamala trees) were active in vitro and in  in vivo against Wolbachia spp. symbionts and Brugia pahangi in gerbils, respectively. Moreover Niclosamide and  Niclosamide ethanolamide also impacted on adults worms by suposedly autophagic inducing effects whereas Niclosamine and Rottlerin affected the fecundity and embryogenesis of the female worms. Collectivelly, all these results are quite clear cut and nicelly summarised in clear comprehensive tables and figures. Consequently I consider that this work contains sufficient and consistent new findings and knowledge to improve the chemotherapy of lymphatic filarial infections in endemic countries and therefore I advise its publication in Pharmaceutics but after only two minor amendements.

1-The gender and species scientific names of parasites, host and remaining living organisms should be typed in italics in references as well (unless otherwise is advised by the Journal rules).

2.- Paragraph 2.5. in  Material an Methods section looks like a bit repetitive and confusing: I suggest steps 2, 3 and 4 be summarised in one as the only variant seems to be the dosage (20,100 or 200 mg/kg), respectively. 

Author Response

1-The gender and species scientific names of parasites, host and remaining living organisms should be typed in italics in references as well (unless otherwise is advised by the Journal rules).

Response: Thank you for the comment. We corrected this.

2.- Paragraph 2.5. in  Material an Methods section looks like a bit repetitive and confusing: I suggest steps 2, 3 and 4 be summarised in one as the only variant seems to be the dosage (20,100 or 200 mg/kg), respectively. 

Response: Thank you for your suggestion. We have revised section 2.5 of the Materials and Methods by summarizing some common aspects of the in vivo experiments. However, we would like to keep the description of each group separately as we believe that it will help the reader to follow the various experiments. We hope the new edits are helpful and clearer. Lines: 167-183